# VISUAL GROUNDING WITH ATTENTION-DRIVEN CONSTRAINT BALANCING

## ABSTRACT

Unlike Object Detection, Visual Grounding task necessitates the detection of an object described by complex free-form language. To simultaneously model such complex semantic and visual representations, recent state-of-the-art studies adopt transformer-based models to fuse features from both modalities, further introducing various modules that modulate visual features to align with the language expressions and eliminate the irrelevant redundant information. However, their loss function, still adopting common Object Detection losses, solely governs the bounding box regression output, failing to fully optimize for the above objectives. To tackle this problem, in this paper, we first analyze the attention mechanisms of transformer-based models. Building upon this, we further propose a novel framework named Attention-Driven Constraint Balancing (**AttBalance**) to optimize the behavior of visual features within language-relevant regions. Extensive experimental results show that our method brings impressive improvements. Specifically, we achieve constant improvements over five different models evaluated on four different benchmarks. Moreover, we attain a new state-of-the-art performance by integrating our method into QRNet.

## 1 INTRODUCTION

Visual grounding (Kazemzadeh et al., 2014; Mao et al., 2016; Plummer et al., 2015; Yu et al., 2016) aims to localize a target object described by a free-form natural language expression, which can be a phrase or a long sentence. It plays a crucial role in many downstream tasks of multi-model reasoning systems, such as visual question answering (Gan et al., 2017; Wang et al., 2020; Zhu et al., 2016) and image captioning (Anderson et al., 2018; Chen et al., 2020; You et al., 2016).

Previous works on visual grounding can be divided into three groups: two-stage method, one-stage method, and transformer-based ones. The two-stage or one-stage methods depend on a complex module with manually-devised techniques to conduct language inference and multi-modal integration. However, TransVG (Deng et al., 2021) proposes a transformer-based method to avoid biases introduced by manual designs, which consists of a DETR (Carion et al., 2020) model to extract visual features, a BERT (Devlin et al., 2018) model for language features extraction, and a transformer encoder to fuse these features along with a learnable object query. Finally, an MLP module processes the object query to obtain the final prediction. Owing to its simplified modeling and superior performance, subsequent studies continue to tap into the potential of this pipeline. These works propose modules to enable interaction between the two modalities before the later fusion or decoding stage, thereby focusing the visual features on areas relevant to language expression, which results in a language-related visual representation that facilitates the subsequent retrieval of the target.

Despite the advancements, the loss functions in these methods solely take into account the regression output of the object query, without providing explicit guidance on how effectively the model concentrates on areas relevant to the language expression. This may make them difficult to optimize for the above aim of extracting language-related visual features, resulting in a suboptimal performance.

In this paper, to fully optimize the alignment between the two modalities from the perspective of the attention behavior, we first investigate the relationship between the attention value of language-modulated visual tokens and the models' performance. We conclude that higher attention values within the ground truth bounding box (bbox) generally indicate better overall performance. However, this phenomenon does not strictly hold universally and exhibits irregular variations across

different models, layers, and datasets. Based on the analysis, we introduce a framework of attention-driven constraint balancing (**AttBalance**) to dynamically impose and balance constraints of the attention during the training process, while also addressing the data imbalance problems that these constraints may cause. Specifically, we propose the Attention Regulation, which consists of the Rho-modulated Attention Constraint to focus the attention of language-modulated visual tokens on the bbox region and the Momentum Rectification Constraint to rectify such harsh guidance, aiming to balance the regulation of the attention behavior (we leave the details of the motivation for each constraint in Sec. 4.2). Furthermore, to balance the influence of different training samples that have varying optimization difficulties due to our constraints, we propose the Difficulty Adaptive Training strategy to dynamically scale up the losses.

To summarize, we have three-fold contributions: **(i)** We unveil the correlation between the attention behavior and the model's performance. **(ii)** We devise an innovative framework, **AttBalance**, to balance the regulation of the attention behavior during training and mitigate the data imbalance problem. Our framework can be seamlessly integrated into different transformer-based methods. **(iii)** Extensive experiments show the performance superiority of our proposed methods.

## 2 RELATED WORK

### 2.1 VISUAL GROUNDING

Visual grounding methods can be roughly classified into three pipelines: two-stage methods, one-stage methods, and transformer-based methods.

**Two-stage methods.** Two-stage approaches (Yu et al., 2018; Chen et al., 2021) treat visual grounding task as the way that firstly generates candidate object proposals and then finds the best matching one to the language. In the first stage, an off-the-shelf detector is used to process the image and propose a set of regions that might contain the target. In the second stage, a ranking network calculates the similarity of the candidate regions and processed language features, and selects the region which has the best score of similarity as the final result. Training losses in this stage include binary classification loss (Plummer et al., 2018) or maximum-margin ranking loss (Yu et al., 2018). In real applications, for better understanding the language expression and the matching of the two modalities, MattNet (Yu et al., 2018) mainly focuses on decomposing language into three components named subject, location, and relationship. Ref-NMS (Chen et al., 2021) introduces an expression-aware score for better ranking the candidate regions. However, this pipeline heavily depends on the pre-trained detector in the first stage, as the second stage will fail to retrieve the correct region if the referred object is not proposed by the first stage. And the features of the two modalities have not been fully integrated.

**One-stage methods.** One-stage approaches (Yang et al., 2019b; 2020) directly concatenate vision and language features in channel dimension and rank the confidence value of candidate regions which is proposed based on concatenated multimodal features. For example, FAOA (Yang et al., 2019b) predicts the bounding box by using a YOLOv3 detector (Redmon & Farhadi, 2018) on the concatenated features. ReSC (Yang et al., 2020) further improves the ability to ground complex queries by introducing a recursive sub-query construction module. However, this pipeline suffers from the trivial concatenation fusion manner of two modalities.

**Transformer-based methods.** Transformer-based approach is first introduced by TransVG (Deng et al., 2021). Different from the above methods, they utilize transformer (Vaswani et al., 2017) encoders to perform cross-modal fusion among a learnable object query, visual tokens, and language tokens. The object query is then processed through an MLP module to predict the bounding box. Benefiting from the flexible structure of transformer modules in processing multi-modalities features, recent works continue to adopt this pipeline and further propose novelties regarding the feature extraction. VLTVG (Yang et al., 2022) comes up with a visual-linguistic verification module before the decoder stage to explicitly encode the relationship between visual and language. QRNet (Ye et al., 2022) proposes a Query-modulated Refinement Network to early fuse visual and language features to alleviate the potential gap between features from a pretrained unimodal visual backbone and features needed to reasoning on both image and language. These methods simply supervise the bounding box prediction result of the object query, modeling the fusion quality of two modalities in an implicit way, which makes the training insufficient.

# 3 ANALYSIS

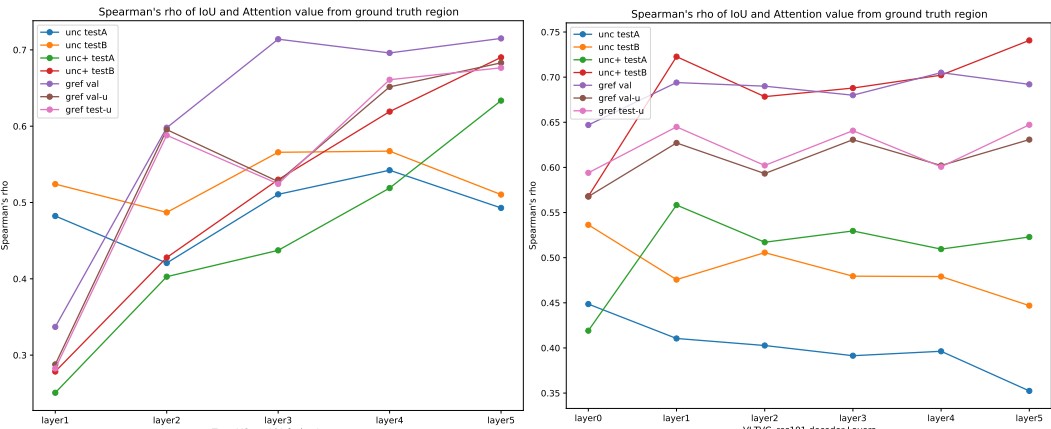

Figure 1: Y-axis represents the Spearman's rank correlation between the performance (IoU) of models (TransVG, VLTVG) and the summation of attention values within the ground truth bounding box across the majority of the evaluation dataset. The X-axis denotes attention derived from different layers. The lines of different colors represent different datasets.

We aim to assess the dependence between the attention behavior of language-modulated visual tokens and the models' performance.[1] Specifically, we sum up the attention value within the ground truth bounding box (bbox) of each fusion or decoding layer from the object query to the visual tokens, indicating the degree of concentration on language-modulated visual tokens within the language-related region. Then we record the IoU value of the corresponding data points. Using Spearman's rho, we analyze the statistical dependence between this attention value and IoU on most of the evaluation datasets from two representative transformer-based models, TransVG and VLTVG.

As shown in Fig. 1 (zooming in), we have three main conclusions: **Conclusion 1)** Since all rho values are positive and the model's predictions depend on the attention behavior, we propose higher attention values within the bbox may indicate better performance. It is an intuitive concept; for precise localization, the model ought to concentrate more on the target area. **Conclusion 2)** As no rho value reaches 1, this positive correlation does not universally hold. This result is reasonable, considering that the language in Visual Grounding often contains background context, which the model requires to infer the foreground. For example, in Fig.2, we need to notice "another dog" outside the bbox to infer "dog at the left" inside the bbox. In these cases, higher attention values within the bbox cannot guarantee better performance. **Conclusion 3)** The correlation degree varies across layers, models, and datasets, with no clear pattern as the depth of the layer increases. This is intuitive, given that there is no predetermined path for the model's decision-making process while dealing with diverse texts and images by considering the varying reasoning capabilities of different models.

# 4 METHOD

In this section, we first review the preliminary for the attention map extraction of transformer-based models in visual grounding Sec. 4.1. Then we provide a detailed explanation of the motivation and implementation for our proposed framework, **AttBalance**, which is composed of two primary elements. The first element is the Attention Regularization, which is further detailed in Sec. 4.2. The second element is Difficulty Adaptive Training, elaborated in Sec. 4.3.

## 4.1 PRELIMINARY

Typically, the transformer-based methods initiate a learnable object query to fuse with visual and language features through self-attention in the fusion stage or cross-attention in the decoding stage,

---

[1]It is after the 0th layer of the fusion module that the visual tokens can be seen as language-modulated visual tokens in the case of TransVG.

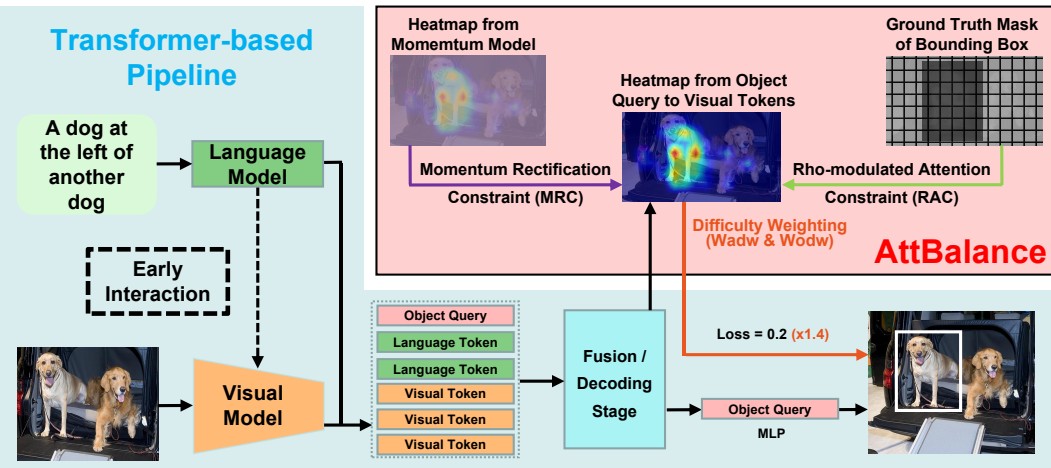

Figure 2: Our AttBalance applied to the transformer-based pipeline. The Early Interaction module may exist in some transformer-based models, e.g., QRNet and VLTVG. The Attention Regularization is formulated as Rho-modulated Attention Constraint and Momentum Rectification Constraint. The Difficulty Weight is used to adaptively scale up the losses to mitigate the data imbalance problem.

and then process this learnable token through an MLP module to regress the bounding box. Our method extracts the attention map between the object query and visual tokens of each layer of the fusion module (as TransVG, QRNet) or decoding module (as VLTVG). To be more specific about the pipeline, take TransVG as an example. As shown in Fig. 2, we first extract the visual and text features from the visual model and the language model, respectively. The visual model is the backbone and the encoder of DETR and the language model is initialized by BERT. Given an image $\mathbf{I} \in \mathbb{R}^{3 \times H_0 \times W_0}$, the visual model generates a 2D feature map followed by a linear projection to reduce the channel dimension as the same as the text features. Therefore, the visual encoded features are extracted and flattened as $\mathbf{z}_v = [\mathbf{p}_v^1, \mathbf{p}_v^2, \dots, \mathbf{p}_v^{N_v}] \in \mathbb{R}^{C \times N_v}$, where $\mathbf{p}_v$ indicates a single visual token, $N_v = \frac{H_0}{32} \times \frac{W_0}{32}, C = 256$. Position embedding is then added to these features to be sensitive to the original 2D location. For the text input, we extract its feature as a sequence of embedding by BERT followed by a linear projection to reduce the channel dimension as the same as the visual encoded feature. Therefore, the encoded text feature is $\mathbf{z}_t = [\mathbf{p}_t^1, \mathbf{p}_t^2, \dots, \mathbf{p}_t^L] \in \mathbb{R}^{C \times L}$, where $\mathbf{p}_t$ indicates a single text token and $L$ is the max expression length. Then $\mathbf{z}_v$ and $\mathbf{z}_t$ are concatenated by inserting a learnable embedding $\mathbf{z}_r$ (*i.e.*, the object query) at the beginning of the concatenated sequence. The whole sequence is formulated as $\mathbf{x}_c = [\mathbf{z}_r, \mathbf{z}_t, \mathbf{z}_v]$. TransVG utilizes many layers of the transformer encoder to fuse the whole sequence and regresses the final bounding box by processing the output object query through an MLP module. In each layer $i$, We treat $\mathbf{z}_{r_i}$ as $\mathbf{Q}_i$ and $\mathbf{z}_{v_i}$ as $\mathbf{K}_i$, and compute the attention map between them.

## 4.2 Attention Regularization

**Rho-modulated Attention Constraint (RAC).** Motivated by **Conclusion 1)**, the RAC aims to strictly supervise the attention map to guide it to totally focus on tokens included in the bbox and completely reduce the attention on tokens outside the bbox. Notably, we compute the mean value of the similarity map over heads before the softmax function by default, so that values outside the bbox will not be squeezed out into a lower value by the softmax function before we get the probability distribution map, which makes the constraint harsher. This stringent constraint accentuates the focus on the language-related region after the visual tokens are modulated by the text tokens, resulting in optimizing the model's integration of multimodal semantics. Specifically, we formulate this constraint into a Binary Cross-Entropy (BCE) loss. As the sum of the values of the entire probability map is 1, we strive to make the sum of the attention values within the bbox as close to 1 as possible, and the sum of the attention values outside the bbox as close to 0 as possible. To take into account **Conclusion 3)**, we propose to calculate rho in each iteration and convert the mean of rho from multiple layers to 1, obtaining the relative rho. This relative rho is then used to scale the

BCE loss of each layer. As shown in Fig. 2, we denote $M$ as a segmentation mask where the value of pixels inside the bbox is assigned as 1 and the value of pixels outside the bbox is assigned as 0. We downsample this mask to the resolution of the visual feature. The RAC, $L_{rac}$, is formulated as below:

$$
\begin{cases}
\text{Attention}_i = \text{Softmax}\left(\text{E}_{\text{h}}\left(\dfrac{\mathbf{Q}_i\mathbf{K}_i^T}{\sqrt{d_k}}\right)\right), \\[2ex]
rho_i^{'} = rho_i - \left(\displaystyle\sum_{i=0}^{n} rho_i/n\right) + 1, \\[3ex]
L_{rac} = \displaystyle\sum_{i=0}^{n} rho_i^{'}\Big(-\log\left(\sum \text{Attention}_i \odot M\right) \\[2ex]
\qquad\qquad - \log\left(1 - \sum \text{Attention}_i \odot \overline{M}\right)\Big),
\end{cases}
\tag{1}
$$

where $i$ is the index of layer, $n$ is the number of layer, $\text{E}_{\text{h}}$ means averaging over heads, $d_k$ is the number of channels, $\odot$ is the Hadamard product and $\overline{M}$ is the inverse of $M$.

**Momentum Rectification Constraint (MRC).** Motivated by **Conclusion 2**), we introduce the Momentum Modal (MomModal), an online self-distillation method proposed by ALBEF (Li et al., 2021). They treat the inaccurate data as noise and use the MomModal, which is updated by taking the moving-average of parameters, to smooth the dramatic change of the learning curve caused by the loss from noise data. Similarly, for situations which the model needs to attend to the background, the training data of the RAC will be treated as noise. Therefore, as shown in Fig. 2, we use the attention map from the MomModal to rectify the constraint from the RAC, a.k.a, Momentum Rectification Constraint (MRC) [2]. The MRC, $L_{mrc}$, is formulated as a KL-divergence:

$$
\begin{cases}
L_{mrc_i} = \text{KL}\left(\text{Attention}_i^{Mom} \,\|\, \text{Attention}_i\right), \\[2ex]
L_{mrc} = \displaystyle\sum_{i=0}^{n} L_{mrc_i},
\end{cases}
\tag{2}
$$

where $\text{Attention}_i^{Mom}$ comes from the MomModal.

We formulate our Attention Regularization as the loss $L_{ar}$, which combines the above two parts, i.e., $L_{ar} = L_{rac} + L_{mrc}$.

## 4.3 DIFFICULTY ADAPTIVE TRAINING (DAT)

Since we introduce an additional loss to the Visual Grounding task, we are also concerned about the imbalance problem brought by $L_{ar}$. As depicted on the left of Fig. 3, we partition the attention values within the ground truth region of VLTVG's last layer into 8 equal number parts, omitting the extreme intervals, i.e. those less than 0.1 or greater than 0.9. The result indicates that the majority of samples are concentrated in intervals with high attention values, leading to an imbalance for our constraint. Specifically, the model may primarily focus on learning these easy cases, as high attention values within bbox lead to low values for $L_{rac}$, which is the main contributor to the $L_{ar}$'s value, as seen from the experimental process. Therefore, in order to direct the model's attention towards the hard cases, we propose the Difficulty Adaptive Training (DAT) strategy, which mainly contributes two weights to dynamically scale up the losses to pay more attention to hard cases related to the difficulty of optimizing the $L_{ar}$. Firstly, we formulate an Actual Difficulty Weight ($W_{adw}$) as below, which is directly proportional to the $L_{ar}$, to indicate the actual difficulty in each iteration for the object query to notice salient features related to the language expressions:

---

[2]Note that there have been many works that leverage the momentum modal to provide supervision, e.g., VL Representation Learning (Li et al., 2021), SSL image classification (Tarvainen & Valpola, 2017), SSL object detection (Liu et al., 2021) and SSL visual grounding (Sun et al., 2023). We, for the first time, show its effectiveness in balancing the constraints on attention behavior in Visual Grounding. Its non-triviality is validated in the ablation study.

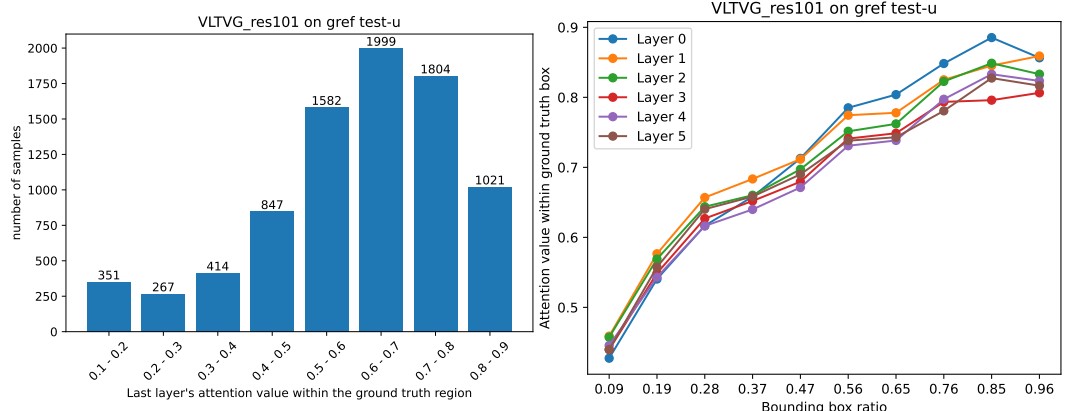

Figure 3: Imbalance study of attention value within ground truth region. **Left**: A histogram analysis of the attention values within the ground truth region of VLTVG's last layer. **Right**: The averaged attention values within the ground truth region of VLTVG's last layer under varying box ratios.

$$\mathrm{W}_{adw} = 0.5 + 1/\left(1 + \exp\left(-L_{ar}\right)\right), \tag{3}$$

where 0.5 ensures the weight remains greater than 1, and $L_{ar}$ is simplified by excluding rho.

Moreover, as shown on the right of Fig. 3, we split the ratios of the bbox to the image size of all data into 10 splits evenly. It can conclude that a higher bbox ratio naturally leads to higher attention value inside the bbox, which means that objectively the optimization for the $L_{ar}$ is harder in cases with a small bbox. Therefore, we scale up the losses based on this factor to solve the imbalance problem, formulating the Objective Difficulty Weight ($W_{odw}$) as below:

$$\mathrm{W}_{odw} = 0.5 + 1/\left(1 + \exp\left(boxRatio - 1\right)\right). \tag{4}$$

where $boxRatio$ is the proportion of bbox to the size of the image.

Therefore, the total loss function is the summation of $L_{ar}$ and the original losses used in Visual Grounding but adaptively adjusted by the DAT:

$$L = \alpha_{ar}L_{ar} + W_{odw}W_{adw}\left(\alpha_1 L_1 + \alpha_g L_{giou}\right), \tag{5}$$

where $\alpha_{ar}$, $\alpha_1$ and $\alpha_g$ are hyperparameters. $L_1$ is the L1 loss and $L_{giou}$ is the GIoU loss (Rezatofighi et al., 2019).

## 5 EXPERIMENTS

### 5.1 DATASET AND EVALUATION

**Dataset.** We evaluate our method on four widely used datasets including RefCOCO (Yu et al., 2016), RefCOCO+ (Yu et al., 2016), RefCOCOg-google (Mao et al., 2016), and RefCOCOg-umd (Mao et al., 2016). Images of these datasets for Visual Grounding are selected from MSCOCO (Lin et al., 2014). RefCOCO (Yu et al., 2016) has 19,994 images with 50,000 referred objects and 142,210 referring expressions. It is officially split into four datasets: training set with 120,624 expressions, validation set with 10,834 expressions, testA set with 5,657 expressions, and testB set with 5,095 expressions. RefCOCO+ (Yu et al., 2016) is a harder benchmark since the language of it is not allowed to include location words but just allowed to contain purely appearance-based descriptions. It has 19,992 images with 141,564 referring expressions for 49,856 referred objects. It is split into four datasets: training set with 120,191 expressions, validation set with 10,758 expressions, testA set with 5,726 expressions and testB set with 4,889 expressions. RefCOCOg (Mao et al., 2016) is also a harder benchmark as it contains a large number of hard cases for its flowery and

Table 1: Performance of our AttBalance applied to transformer-based models.

| Models | RefCOCO | | | RefCOCO+ | | | RefCOCOg-g | RefCOCOg-umd | |
| | val | testA | testB | val | testA | testB | val | val | test |
|---|---|---|---|---|---|---|---|---|---|
| *Two-stage:* | | | | | | | | | |
| CMN (Hu et al., 2017) | - | 71.03 | 65.77 | - | 54.32 | 47.76 | 57.47 | - | - |
| VC (Zhang et al., 2018) | - | 73.33 | 67.44 | - | 58.40 | 53.18 | 62.30 | - | - |
| ParalAttn (Zhuang et al., 2018) | - | 75.31 | 65.52 | - | 61.34 | 50.86 | 58.03 | - | - |
| MAttNet (Yu et al., 2018) | 76.65 | 81.14 | 69.99 | 65.33 | 71.62 | 56.02 | - | 66.58 | 67.27 |
| LGRANs (Wang et al., 2019) | - | 76.60 | 66.40 | - | 64.00 | 53.40 | 61.78 | - | - |
| DGA (Yang et al., 2019a) | - | 78.42 | 65.53 | - | 69.07 | 51.99 | - | - | 63.28 |
| RvG-Tree (Plummer et al., 2018) | 75.06 | 78.61 | 69.85 | 63.51 | 67.45 | 56.66 | - | 66.95 | 66.51 |
| NMTree (Liu et al., 2019) | 76.41 | 81.21 | 70.09 | 66.46 | 72.02 | 57.52 | 64.62 | 65.87 | 66.44 |
| Ref-NMS (Chen et al., 2021) | 80.70 | 84.00 | 76.04 | 68.25 | 73.68 | 59.42 | - | 70.55 | 70.62 |
| *One-stage:* | | | | | | | | | |
| SSG (Chen et al., 2018) | - | 76.51 | 67.50 | - | 62.14 | 49.27 | 47.47 | 58.80 | - |
| FAOA (Yang et al., 2019b) | 72.54 | 74.35 | 68.50 | 56.81 | 60.23 | 49.60 | 56.12 | 61.33 | 60.36 |
| RCCF (Liao et al., 2020) | - | 81.06 | 71.85 | - | 70.35 | 56.32 | - | - | 65.73 |
| ReSC-Large (Yang et al., 2020) | 77.63 | 80.45 | 72.30 | 63.59 | 68.36 | 56.81 | 63.12 | 67.30 | 67.20 |
| LBYL-Net (Huang et al., 2021) | 79.67 | 82.91 | 74.15 | 68.64 | 73.38 | 59.49 | 62.70 | - | - |
| *Transformer-based:* | | | | | | | | | |
| TransVG_ResNet50 | 80.49 | 83.28 | 75.24 | 66.39 | 70.55 | 57.66 | 66.35 | 67.93 | 67.44 |
| +AttBalance | **82.90** | **85.87** | **77.69** | **70.84** | **75.96** | **61.63** | **70.61** | **73.69** | **72.44** |
| | **+2.41** | **+2.59** | **+2.45** | **+4.45** | **+5.41** | **+3.97** | **+4.26** | **+5.76** | **+5.00** |
| TransVG_ResNet101 | 80.83 | 83.38 | 76.94 | 68.00 | 72.46 | 59.24 | 68.03 | 68.71 | 67.98 |
| +AttBalance | **82.52** | **85.12** | **78.55** | **71.41** | **75.65** | **62.65** | **70.32** | **74.16** | **72.79** |
| | **+1.69** | **+1.74** | **+1.61** | **+3.41** | **+3.19** | **+3.41** | **+2.29** | **+5.45** | **+4.81** |
| VLTVG_ResNet50 | 84.53 | 87.69 | 79.22 | 73.60 | 78.37 | 64.53 | 72.53 | 74.90 | 73.88 |
| +AttBalance | **85.30** | **88.13** | **81.50** | **74.86** | **80.21** | **64.68** | **74.19** | **76.65** | **74.89** |
| | **+0.77** | **+0.44** | **+2.28** | **+1.26** | **+1.84** | **+0.15** | **+1.66** | **+1.75** | **+1.01** |
| VLTVG_ResNet101 | 84.77 | 87.24 | 80.49 | 74.19 | 78.93 | 65.17 | 72.98 | 76.04 | 74.18 |
| +AttBalance | **85.30** | **88.13** | **81.50** | **75.14** | **80.25** | **66.34** | **74.08** | **77.35** | **75.61** |
| | **+0.53** | **+0.89** | **+1.01** | **+0.95** | **+1.32** | **+1.17** | **+1.1** | **+1.31** | **+1.43** |
| QRNet | 84.01 | 85.85 | 82.34 | 72.94 | 76.17 | 63.81 | 71.89 | 73.03 | 72.52 |
| +AttBalance | **87.32** | **89.64** | **83.87** | **77.51** | **82.03** | **68.64** | **77.40** | **79.86** | **79.63** |
| | **+3.31** | **+3.79** | **+1.53** | **+4.57** | **+5.86** | **+4.83** | **+5.51** | **+6.83** | **+7.11** |

complex expressions. Especially, the length of language expression regarding RefCOCOg is much longer than that of other datasets. It has 25,799 images with 49,822 object instances and 95,010 expressions. There are two commonly used splitting conventions. One is RefCOCOg-google (Mao et al., 2016) with a training set and a validation set, and the other is RefCOCOg-umd (Nagaraja et al., 2016) with a training set, a validation set, and a test set.

**Evaluation.** Following the previous setting in the previous work (Deng et al., 2021; Yang et al., 2022), we use the top-1 accuracy(%) to evaluate our method, where the predicted bounding box will be regarded as positive if its IoU with the ground-truth bounding box is greater than 0.5.

## 5.2 IMPLEMENTATION DETAILS

We verify the pluggability and effectiveness of our proposed AttBalance by applying it to five models: one baseline model (TransVG) with two kinds of backbone (ResNet50 & ResNet101), one state-of-the-art model (VLTVG) with two kinds of backbone (ResNet50 & ResNet101), and one state-of-the-art model (QRNet) with one kind of backbone (Swin-S). The details for the constraint of each model are included in the appendix.

In terms of other configurations, we adhere to the original setup employed by TransVG, VLTVG, and QRNet, respectively. However, the implementation of RandomSizeCrop augmentation in TransVG and QRNet seriously cuts off the ground truth region, causing our $L_{ar}$ to be greatly affected. Thus, we constrain it to save the ground truth region. We set $\alpha_{ar} = 1$, $\alpha_1 = 1$, $\alpha_g = 1$. The momentum parameter for updating the momentum model is set as 0.9.

## 5.3 QUANTITATIVE RESULTS

As presented in Table 1, we jointly analyze them and can easily draw the following observations: **(i)** Under the guidance of our AttBalance, all transformer-based models consistently obtain an impressive improvement on all benchmarks. TransVG(+AttBalance) achieves an average improvement of 3.55% across all benchmarks; VLTVG(+AttBalance) achieves an average improvement of 1.16%

across all benchmarks; QRNet(+AttBalance) achieves an average improvement of 4.82% across all benchmarks. These results indicate that our AttBalance is a general constraint framework which can be seamlessly transferred to existing methods and brings a notable improvement. **(ii)** To the best of our knowledge, QRNet(+AttBalance) achieves a new state-of-the-art performance in the Visual Grounding task, excluding those pretraining works. **(iii)** Compared to RefCOCO, RefCOCOg-google and RefCOCOg-umd are harder benchmarks since the language in them is much longer, with more flowery and complex expressions. However, the improvements brought by our AttBalance on them are even more remarkable. Specifically, QRNet(+AttBalance) achieves 5.51%, 6.83%, and 7.11% absolute improvement on gref val, gref val-u, and gref test-u, respectively. The same situation happens on other hard benchmarks, like RefCOCO+, whose language is not allowed to include location words and is restricted to purely appearance-based descriptions. We bring 4.45%, 5.41%, and 3.97% absolute improvements on val, testA, and testB, respectively, to TransVG_Res50. **(iv)** The improvements on VLTVG are moderate, likely due to the lack of further interaction between language and visual tokens in the decoding stage. This leads to limited word-level semantics in visual tokens, restricting our guidance in language-related regions.

## 5.4 ABLATION STUDY

In this section, to verify the effectiveness of each module, we conduct comprehensive ablation studies based on TransVG. Specifically, we add the modules we proposed through controlling variates.

Table 2: Ablation study on Momentum Rectification Constraint (MRC), Rho-modulated Attention Constraint (RAC), and Difficulty Adaptive Training (DAT). The blue downward arrow represents a decrease, while the red upward arrow represents an increase. 'Ori' denotes the performance cited from the original paper, and 'Rep' signifies the reproduced performance.

| Ori | Rep | MRC | RAC | DAT | gref val-u | gref test-u | unc+ test-A | unc+ test-B |
|-----|-----|-----|-----|-----|-----------|-------------|-------------|-------------|
| ✓ | | | | | 67.93 | 67.44 | 70.55 | 57.66 |
| | ✓ | | | | 67.77 | 67.52 | 69.08 | 56.42 |
| | ✓ | ✓ | | | 67.63▼ | 67.35▼ | 69.12 | 55.80▼ |
| | ✓ | | ✓ | | 71.81▲ | 71.04▲ | 75.20▲ | 60.42▲ |
| | ✓ | ✓ | ✓ | | 73.35▲ | 72.29▲ | 75.60▲ | 60.71▲ |
| | ✓ | ✓ | ✓ | ✓ | 73.69▲ | 72.44▲ | 75.96▲ | 61.63▲ |

As shown in Table 2, Ori refers to the performance reported in the original paper. Since we slightly change the augmentation, RandomSizeCrop, we also reproduce the performance influenced by this factor, denoted as Rep. Based on the table, it is shown that our modification to RandomSizeCrop does not yield significant changes in the model's performance. The introduction of the RAC module leads to a performance improvement. Conversely, the incorporation of only the MRC module results in a decline on average, suggesting that merely smoothing the attention behavior by MomModal does not bring benefits. This might be due to the fact that the previous models ensembled in MomModal are still more underfitting than the current one; therefore, the guidance from their attention map is not consistently reliable. However, only when the MRC is used to rectify the RAC can it lead to an increase, since the potential bias of focusing attention on language-related regions is smoothed. Furthermore, the utilization of DAT yields an improvement as well.

Table 3: Ablation study on the relative rho factor of the Rho-modulated Attention Constraint (RAC).

| TransVG_Res50 | gref val | gref val-u | gref test-u | unc+ testA | unc+ testB |
|---------------|----------|-----------|-------------|------------|------------|
| AttBalance w/o rho | 70.46 | 73.49 | 72.39 | 75.83 | 60.05 |
| AttBalance | 70.61 | 73.69 | 72.44 | 75.96 | 61.63 |

Since our RAC module incorporates rho to consider **Conclusion 3)**, we also report the effectiveness of rho for our AttBalance. Table 3 indicates that rho can consistently lead to improvements.

Table 4: Ablation study regarding the constraint on the number of layers in the TransVG. "1 layer" refers to the last layer, and "2 layers" refers to the last two layers, and so on.

| TransVG_Res101 | 1 layer | 2 layers | 3 layers | 4 layers | 5 layers |
|----------------|---------|----------|----------|----------|----------|
| val-u | 74.18 | 73.63 | 73.73 | 74.16 | 72.71 |
| test-u | 72.44 | 72.80 | 72.69 | 72.79 | 71.11 |

As shown in Table 4, We also report an ablation study concerning the application of our AttBalance to the number of layers in TransVG, ultimately choosing to apply it to the last four layers. Since the

visual tokens in the 0th layer have not yet interacted with the language tokens, they are not suitable for our AttBalance.

## 5.5 SEMI-SUPERVISION STUDY

Semi-supervision for Visual Grounding is dedicated to addressing the situation of scarce labels. The current state-of-the-art method (Sun et al., 2023) trains on a small number of labels and then generates pseudo labels of the remaining unlabeled data for continued training. We conduct training of TransVG(+AttBalance) only on the small amount of labels, without additional pseudo label training, in comparison with the SOTA semi-supervised method.

Table 5: Comparison on semi-supervision of Visual Grounding.

| 10% label | unc val | unc+ val | gref val-u |
|---|---|---|---|
| TransVG_Sup (Sun et al., 2023) | 67.2 | 43.7 | 47.9 |
| TransVG_ReT (Sun et al., 2023) | 70.3 | 46.4 | 51.0 |
| TransVG(+AttBalance) | 71.86 | 51.00 | 58.37 |

As shown in Table 5, TransVG_Sup denotes the baseline performance with supervised learning on 10% of the labels; TransVG_ReT represents the state-of-the-art semi-supervised approach, utilizing 10% of the labels for supervised learning and the remaining 90% for pseudo label learning; TransVG(+AttBalance) signifies our method, which only conducts supervised learning on 10% of the labels. Our method significantly outperforms the baseline performance in a semi-supervised setting, even exceeding the performance of the state-of-the-art semi-supervised method by 7.37% on gref val-u and 4.6% on unc+ val, despite utilizing 90% fewer unlabeled data.

## 5.6 QUALITATIVE RESULTS

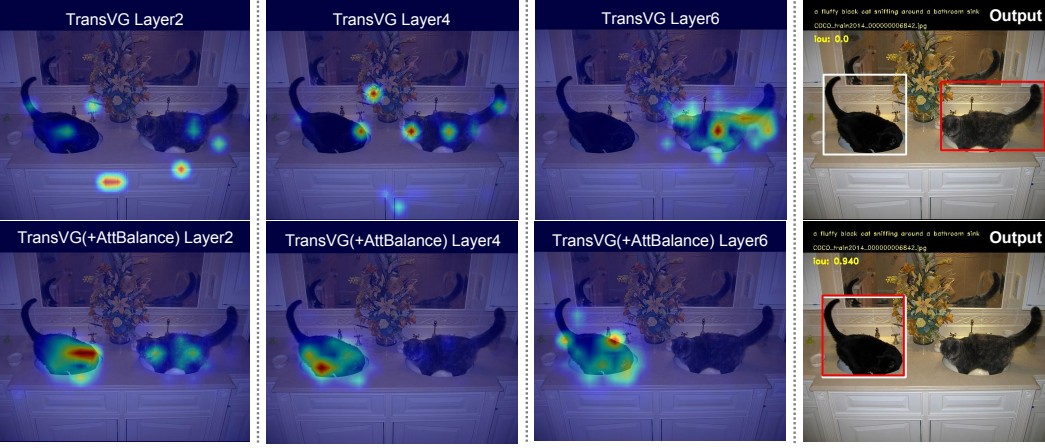

Figure 4: "a fluffy black cat sniffing around a bathroom sink". We visualize the attention map of each layer. The white box is the ground truth and the red box is the prediction.

As presented in Fig.5, we visualize the attention map in the 2nd, 4th, and 6th fusion layers of TransVG and TransVG(+AttBalance). This scenario involves two black cats in a bathroom sink, one of which is sniffing. This hard case requires models to fully exploit the connection of the text and the image to locate the left "black cat in a bathroom sink" under reasoning on "sniffing", even if the right one is also black and in a bathroom sink. The result of TransVG cannot locate the "cat" in the early layer, and finally fails to judge which one is "sniffing". In contrast, under the constraint of our AttBalance, the model can quickly locate two "black cat" candidates only at layer2, and correctly infer the left one only at layer4.

## 6 CONCLUSION

In this paper, we first analyze the correlation between attention behavior and the model's performance. Based on this analysis, we propose a framework, named AttBalance, to incorporate language-related region guidance for fully optimized training. Specifically, the framework consists of Attention Regularization, which balances the constraints on the attention behavior, and the Difficulty Adaptive Training strategy to mitigate the imbalance problem of these constraints.

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

# A   APPENDIX

**Implementation Details.** Our AttBalance is applied to the final four layers of TransVG for both backbone models during the first 60 epochs. For QRNet, we apply our AttBalance to all six layers for the first 60 epochs. For VLTVG, there are two settings for our AttBalance. One is constraining all six layers of our modified attention map throughout the entire 90 epochs. The other is constraining the normal attention map of the last layer for the first 60 epochs, then only retaining the box loss of the final layer's output. We experimentally select the setting with the best performance. We set the batch size to 64 for all models, except for QRNet where we set it to 56 due to its high GPU memory requirements.

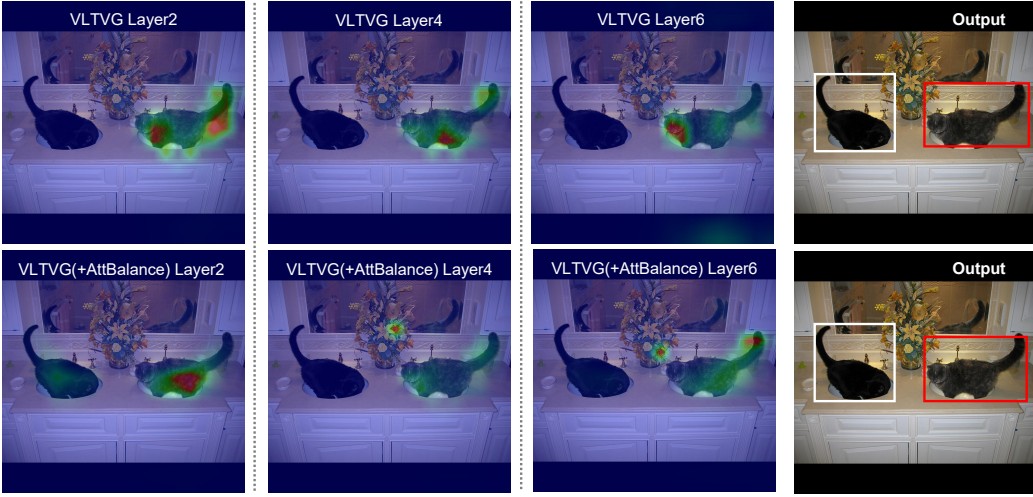

Figure 5: "a fluffy black cat sniffing around a bathroom sink". We visualize the attention map of each layer. The white box is the ground truth and the red box is the prediction.

In Fig. 5, we visualize the attention map of VLTVG(+AttBalance). Here, we can see that the attention of VLTVG(+AttBalance) does shift a little bit compared to the original one, but it is subtle, resulting in a wrong prediction. Such qualitative results align with the fact that the improvement on VLTVG by AttBalance is more subtle than those on TransVG and QRNet.

