# OpenReview forum: "Visual Grounding with attention-driven constraint balancing"
_ICLR.cc/2024/Conference — Submitted to ICLR 2024_

### Official Review · Reviewer_qRXz · 2023-10-30

**Soundness:** 3 good
**Presentation:** 3 good
**Contribution:** 3 good
**Rating:** 6
**Confidence:** 3

**Summary:**

This paper focuses on improving Visual Grounding using Transformer-based models. They first analyze the correlation between the model fusion/decoder layer's attention score within the ground truth bounding box and the corresponding IoU value for multiple benchmarks. Based on their observations, they propose two objectives: (1) Rho-modulated Attention Constraint (RAC); and (2)
Momentum Rectification Constraint (MRC) for attention regularization. RAC constraints the model to generate attention maps focusing on the text included in the bounding box. MRC is a momentum distillation module for rectifying RAC in cases in which background information is needed. They also introduce Difficulty Adaptive Training to make the model pay more attention to hard samples. Finally, they conduct experiments using multiple transformer-based models on several benchmarks including RefCOCO, RefCOCO+, and RefCOCO-g. Their experimental results show better performances than the baselines.

**Strengths:**

(1) They design the objectives from the analysis and observations, which makes the design easy to follow and intuitive.
(2) The proposed RAC, MRC loss, and the DAT training strategy can be adapted to different transformer-based visual grounding models, therefore the methods are general to use in the future.
(3) They conduct comprehensive ablation experiments to show each proposed element is needed to get the best performances on multiple benchmarks. Also, their main results show consistent improvements over these benchmarks using different backbone models.

**Weaknesses:**

(1) The benchmarks used in the paper are RefCOCO, RefCOCO+ and RefCOCO-g. But I see in the supplementary material the authors also try ReferItGame and Flicker30K. Could you also provide additional experimental results on these benchmarks to compare with the baselines?
(2) By adding the RAC and MRC, how long did you train the model? Can you compare it with the baseline training time as well?

**Questions:**

Please answer the questions in Weaknesses.

---

> ### Author Response · Authors · 2023-11-18
>
> # Response 4.1: Response to Weakness 1 about additional experimental results.
> Thank you for your inquiry regarding additional experimental results. First, we would like to clarify the reason we did not conduct experiments on ReferItGame and Flicker30K in our paper. However, we are pleased to provide the additional experiment results you requested on ReferItGame. Due to the enormity of the Flicker30K training set and the limited time and training resources during the rebuttal period, we cannot guarantee the completion of all experiments during the rebuttal phase. However, we will include all experiments in the final version's appendix.
>
> **Reasons for omitting experiments on ReferItGame and Flicker30K:** Flicker30K is primarily a phrase grounding benchmark, and it may not fully represent the challenges of Visual Grounding characterized by complex and free-form language expression. This distinction is evident in MDETR [1], where Table 2 in their paper outlines benchmarks for Visual Grounding (also known as referring expression comprehension) using RefCOCO, RefCOCO+, and RefCOCOg, akin to our approach. However, Table 3 in their paper is dedicated to the phrase grounding task using Flicker30K. Similar configurations are observed in SiRi, where validation is conducted solely on RefCOCO, RefCOCO+, and RefCOCOg. Considering the scope of our experiment, which involves evaluating five models on four benchmarks, resulting in 45 quantitative results and numerous ablation studies, additional validation on ReferItGame and Flicker30K would extend beyond the intended experiment scope. Specifically, Flicker30K comprises over 427k training samples.
>
>
> **Additional experiments on ReferItGame:** Nevertheless, to address your concerns to a certain extent, we performed an additional experiment on ReferItGame. Here, we reproduced the TransVG model and conducted an experiment with TransVG(+AttBalance). The results, presented in the table below, demonstrate that our AttBalance on ReferItGame indeed results in a significant improvement.
> |                | TransVG | TransVG(+AttBalance) |
> |----------------|---------|-----------------------|
> | ReferItGame val | 71.94   | 73.84                 |
> | ReferItGame test| 69.69   | 70.57                 |
>
> [1] MDETR -- Modulated Detection for End-to-End Multi-Modal Understanding. ICCV 2021.
>
> # Response 4.2: Response to Weakness 2 about training time.
> Thank you for your question regarding the training time for our model with the inclusion of RAC and MRC compared to the baseline. We appreciate your interest in this aspect of our work. To address your question, we investigate the training time consumption of both the baseline (TransVG) and our method (TransVG(+AttBalance)) using the experiment results from the table provided earlier. It is essential to note that all experiments were conducted on 8 Quadro RTX 6000 GPUs with the same CUDA and PyTorch environment. As demonstrated in the table below, the additional training time introduced by our AttBalance falls within a reasonable range.
>
> |               | TransVG          | TransVG(+AttBalance) |
> |---------------|------------------|-----------------------|
> | Training time | 11 hours 05 mins | 13 hours 53 mins      |

---

### Official Review · Reviewer_zMiJ · 2023-10-31

**Soundness:** 2 fair
**Presentation:** 3 good
**Contribution:** 2 fair
**Rating:** 5
**Confidence:** 4

**Summary:**

This study first pinpoints a general problem that the loss function adopted in visual grounding tasks does not differ from that in object detection tasks, failing to consider the alignment between the language expressions and the visual features. To that end, the authors propose an attention-driven constraint balancing (AttBalance) module to explicitly constrain attention to focus more in the range of the bounding box. Experimental results are presented on various datasets using different models to validate the effectiveness of AttBalance.

**Strengths:**

- The authors point out an important problem that loss functions in visual grounding is not specially designed to consider vision-language interactions.

- An attention balance method is proposed based on the found positive correlation between the attention value inside a bounding box and the model’s performance.

- The proposed module is able to achieve performance gain across different methods on major visual grounding tasks.

**Weaknesses:**

- My major concern is about the intuitive motivation of this study. The authors argue that “higher attention values within the ground truth bounding box (bbox) generally indicate a better overall performance” through two individual experiments. Specifically, a Spearman’s rank correlation between the attention values and the models’s predicted IoU is shown to indicate the positive correlation between the model’s prediction and the attention value. Even though the results using TransVG-R101 generally comply with the assumption, the results using VLTVG-R101 barely do so. The trend on unc testA even contradicts the argument.

- Moreover, one could list a lot of cases where the attention values inside a bounding box do not necessarily correlate with the models' performance. Take small objects for example, the most influential factors would lie in the background for determining the semantics of the object instead of the attention inside the bounding box. Therefore, the motivation of this study fails to persuade the reviewer of the effectiveness of the proposed module in various scenarios.

- The introduced regulation on the attention value is to guide the attention to focus more on the area inside a bounding box. Even though the authors assert that the loss could apply to different architectures, more hyperparameters are also introduced which would also hurt its transferability on other visual grounding models. For example, in the sentence “we partition the attention values within the ground truth region of VLTVG’s last layer into 8 equal number parts, omitting the extreme intervals, i.e. those less than 0.1 or greater than 0.9”, one would identify at least 3 hyperparameters whose ablation is also missing in the manuscript.

- Some major writing issues. Each figure annotation should explain itself whereas the annotations in this study e.g., Figure 1 and Figure 3 to too short to comprehend without referring to the main manuscript.

- Minor writing issues, e.g., “analyzing” -> “analyze” in 4.3. The authors are encouraged to further proofread the manuscript.

**Questions:**

The major concerns would be how to prove its motivation in general. I would like to also raise some other question regarding the experimental details.

- In regards to the ablation study, it is weird that only applying MRC even causes degradation of the performance. In contrast, it is able to largely benefit the model when combined with RAC. This is an interesting yet strange phenomenon, especially given that the feature map distillation from a moving-averaged teacher has been effective on various self-supervised/semi-supervised task. This is not well addressed in the manuscript with only one sentence “ “suggesting that merely smoothing the attention behavior by MomModal does not bring benefits”

---

> ### Author Response · Authors · 2023-11-18
>
> # Response 3.1: Response to Weakness 1 about the Spearman’s rank correlation.
> Certainly, we appreciate your attention to Spearman's rank correlation results, particularly in the case of VLTVG-R101 on unc testA. We would like to provide clarification on a few things:
>
> * Spearman's rank correlation (rho) measures the statistical dependence between the rankings of two variables. A positive value of rho indicates the positive association between the ranks of the two variables. In the case of VLTVG-R101 on unc testA, even though the results show a minimum correlation of 0.35, this still indicates a positive correlation. Therefore, it aligns with the conclusions drawn in our ANALYSIS section, specifically Conclusions 1 and 2, stating that higher attention values inside the bounding box generally imply better performance. It is important to note that this is a general trend and not an absolute rule for every data sample.
>
> * Furthermore, the observed variations in correlation degrees across layers, models, and datasets, as indicated by the different trends in each line, support our Conclusion 3. This conclusion highlights that the correlation degree is not uniform and follows no predefined pattern. Given these observations, we propose adapting the scaling of the RAC using Spearman's rank correlation (rho) factor. This adaptive scaling approach aims to account for the variations observed in correlation degrees across different scenarios, eliminating the need for manually designed hyperparameters.
>
> # Response 3.2: Response to Weakness 2 about the motive regarding Attention variability across different scenarios.
> I appreciate your thoughtful consideration and concern regarding the correlation between attention values inside a bounding box and model performance, especially in scenarios involving small objects. Your observation raises a valid point, and we would like to provide further clarification:
>
> * Detecting a target object intuitively involves the model attending to the region of the object. However, as you rightly pointed out, background factors can also significantly influence the final prediction, especially in cases involving small objects. Determining whether the foreground or background matters more is challenging, particularly in visual grounding scenarios where the text is complex and contains many background clues. This nuanced understanding aligns with the findings from our ANALYSIS, where all Spearman's rank correlation (rho) values are positive, but none reach 1.
>
> * This is exactly why we should propose MRC to rectify RAC, where RAC serves as the intuitive motivation, while MRC is introduced to rectify this approach and mitigate excessive bias assumptions. Our motivation comes from the ANALYSIS and is verified in the ablation study. As shown in Table 2, RAC can bring impressive improvement, and MRC further enhances these results.
>
> # Response 3.3: Response to Weakness 3 about the hyperparameters.
> Thank you for your detailed examination of our implementation, particularly concerning hyperparameters. We would like to address the raised concern and maybe some misunderstanding in Figure 3:
>
> **hyperparameters:** As emphasized in Response 2.3 to Reviewer L37Q, we have thoroughly explained the rationale behind these hyperparameters in our answer to that question, and I would encourage you to refer to it for more clarity. Here, for a brief recap, it is crucial to recognize the substantial differences in the models' structure, training settings, and dataset features. The distinct fusion/decoding modules of various models naturally result in slight adaptations of our AttBalance. Furthermore, variations in training settings and dataset characteristics necessitate minor adjustments to the training epoch in our AttBalance.
>
> **Clarify misunderstandings in Figure 3:** In reference to the configuration in Figure 3, it appears there might be a misunderstanding. This setup is used exclusively for analyzing the last layer attention distribution of VLTVG across the dataset. It serves to provide insight into our motivation for incorporating DAT. It is important to note that these are not hyperparameters in our DAT, which is computed adaptively based on BoxRatio and attention regulation for each data sample, without manual-craft design.
>
> We are grateful for the acknowledgment of our generalization and effectiveness, particularly highlighted in the reviews: Strength 2 from Reviewer 6bMP, Strength 1 from Reviewer L37Q, Strength 3 from you, and Strengths 2 & 3 from Reviewer qRXz. It is rewarding to see recognition for our consistent improvement across all five models on four benchmarks.

---

> > ### Author Response · Authors · 2023-11-18
> >
> > # Response 3.4: Response to Weaknesses 4&5 about the writing suggestions.
> > Thank you for your valuable feedback. I appreciate your attention to the minor writing issue, specifically, Figures 1&3 and “analyzing”. We have promptly corrected this error and conducted a thorough proofreading of the entire manuscript to address similar issues and enhance the overall writing quality in the attached revised paper.
> >
> > # Response 3.5: Response to Question about the experimental details related to MRC.
> > Thank you for noting the intriguing observation regarding the impact of applying MRC and RAC on performance. A similar question was addressed in Response 1.3 to Reviewer 6bMP, where we thoroughly explained the underlying rationale. I recommend referring to that response for a more detailed explanation. Here, we provide a brief overview:
> >
> > * In essence, the Momentum model operates as a continuously-ensembled version, incorporating exponentially moving averaged versions from previous training steps. Given that these earlier models are more underfitting than the current one, relying solely on their ensemble can lead to a decline in the reliability of guiding the current model's attention map. However, the Momentum model proves beneficial while serving as a counterpart constraint to smooth the original constraint. In scenarios where the original constraint may not be consistently reliable, such as with the Image-Text Contrastive Learning constraint in ALBEF [1] facing noise unpaired data, the guidance from the Momentum model helps bring the current model's learning target closer to its previous ensemble behavior, resulting in smoother learning.
> >
> > * In the context of visual grounding, RAC effectively directs the model's attention toward language-related regions, generally resulting in improvements, as outlined in conclusion (1) of our ANALYSIS section and illustrated in Table 2 of our ablation study. However, RAC is not a universally reliable constraint, as visual grounding also necessitates considering background objects as potential clues mentioned in the language. Therefore, the consistent benefit observed by using MRC to rectify RAC is discussed in conclusions 2&3 of our ANALYSIS section and validated by Table 2 in our ablation study.
> >
> > To aid future readers' comprehension, we have concisely incorporated this discussion into the revised version of the paper.
> >
> > [1] Align before Fuse: Vision and Language Representation Learning with Momentum Distillation. NeurIPS 2021 Spotlight.

---

> ### Author Response · Authors · 2023-11-20
> **Reminder: Seeking Your Feedback**
>
> Dear Reviewer zMiJ,
>
> We deeply appreciate the time and expertise you have dedicated to reviewing our submission. Your insights are invaluable, and we are thankful for the thorough consideration you have given to our work.
>
> Having diligently responded to and addressed your concerns regarding rho, attention, writing, and the MRC-related experiments, we believe your feedback is pivotal in elevating the quality of our submission.
>
> As we approach the impending deadline, we respectfully urge you to provide your timely response. Your expert perspective is essential to our revision, and your prompt feedback would be immensely beneficial.
>
> Looking forward to hearing from you promptly.
>
> Best regards,

---

### Official Review · Reviewer_L37Q · 2023-11-01

**Soundness:** 2 fair
**Presentation:** 2 fair
**Contribution:** 2 fair
**Rating:** 5
**Confidence:** 4

**Summary:**

The paper tackles the problem of visual grounding, aiming to produce relevant object bounding boxes in the image based on free-form text input. The proposed method imposes explicit constraints to ensure that self-attention within the transformer layers are focused on the ground truth bounding box areas during training. Specifically, the authors introduce (1) Rho-modulated Attention Constraint (RAC), a BCE loss that promotes the sum of the attention values within the ground-truth bounding box to be 1 and 0 elsewhere, (2) Momentum Rectification Constraint (MRC) to help the model converge smoothly with RAC, and (3) Difficulty Adaptive Training (DAT) to dynamically change the weight of losses. When combined with existing models. The boost is large for QRNet and TransVG, but marginal for VLTVG.

**Strengths:**

S1. The proposed approach consistently enhances the performance of existing models when integrated. Notably, when paired with QRNet, it outperforms all other methods that are compared in the paper. However, the paper lacks comparison with more recent SOTA methods such as VG-LAW [a].

[a] Language Adaptive Weight Generation for Multi-task Visual Grounding, Su et al., CVPR 2023

S2. The paper includes an ablation study to assess the impact of the various components proposed in the paper.

S3. The paper provides an analysis of correlation between the grounding performance (IoU of the predicted bounding boxes against the ground truth bounding boxes) and the summation of the attention value within the ground truth bounding boxes that motivates the proposed method.

S4. The paper provides qualitative results, showing the attention maps both with and without the proposed method, which I found to be insightful.

**Weaknesses:**

W1. The paper assumes that the lack of explicit attention guidance results in suboptimal performance. However, it is difficult for me to buy the assumption, given that the model is trained in an end-to-end manner. Factors like the size and diversity of the training dataset could also be responsible if the attention appears dispersed.

W2. The constraints proposed in the paper involve numerous hyperparameters and heuristics, including adding constraints and subsequently introducing other ones to temper the initial constraints.

W3. The discussion of the prior work that uses object detection losses only, which is one of the motivations of the paper, references papers published over two years ago. It might be worth considering more recent work such as [a] that leverages focal loss and a segmentation loss (DICE loss) similar to Segment Anything Model (SAM) and [b] which uses both object-text and patch-text alignment losses. It might be interesting to compare the attention maps with [a] as well, to validate whether there's a genuine need for explicit guidance on attentions.

[a] Language Adaptive Weight Generation for Multi-task Visual Grounding, Su et al., CVPR 2023
[b] YORO - Lightweight End to End Visual Grounding, Ho et al., ECCV 2022

W4. The third contribution highlighted in the paper, namely,  “(iii) Our framework can be seamlessly integrated into different transformer-based methods.” is not a valid contribution, but a feature of the proposed framework mentioned in (ii). I suggest merging it with the contribution (ii).

W5. Presentation: Both the main text and the captions omit explanations for legends and notations in the figures and tables (Figure 1, Table 2). The quality of writing could also be further improved.

[Minor comments]
I suggest revising the abstract for enhanced clarity.

In “Specifically, we achieve uniform gains across five different models evaluated on four different benchmarks.”, I suggest “constantly improves over” rather than “uniform gains”, because the gains are not equal.

Also, references can be adjusted. Instead of “TransVG Deng et al. (2021)”, consider using “TransVG proposed by Deng et al. (2021)” or  “TransVG (Deng et al., 2021)”.

**Questions:**

Q1. The performance improvement is significant for QRNet and TransVG, yet only slight for VLTVG. I'm curious if this variation is mirrored in the attention map. When examining the attention map, similar to what's shown in Figure 4, is the shift in attention for VLTVG more subtle compared to that of QRNet?

Q2. In Section 2’s analysis, I want to double check if the “IOU value” in “Then we record the IoU value of corresponding data points” is the IoU between the object bounding box predictions against the GT bounding boxes.

Q3. In Section 2’s analysis, were the attention values normalized in any way?

Q4. The paper delves into directing visual features towards specific regions (bbox regions) in an image. Instead of their proposed method, why not introduce a learnable 2D regional weighting layer, akin to the approach in [a], to modulate the attention? This would be done as Attention_i = weighted_2D_mask * Attention_i, where the weighted_2D_mask is produced by a shallow convnet that takes a CxWxH feature map and is trained end-to-end without extra supervision. This could serve as another baseline to validate whether there's a need for explicit guidance/supervision on attentions.

[c] Large-Scale Image Retrieval with Attentive Deep Local Features, ICCV 2017

---

> ### Author Response · Authors · 2023-11-18
>
> # Response 2.1: Response to slight inaccuracy in the summary.
> Thanks for your clear summary of our method. There is a slight inaccuracy in the summary; it should be the cross-attention from the object query to vision tokens, instead of self-attention.
>
> # Response 2.2: Response to Weakness 1 about our assumption, considering attention dispersion.
> We appreciate your insightful question about the potential impact causing attention dispersion. We will address this issue both theoretically and based on practical experimental results below.
>
> **Assumption on the behavior of attention:**
>
> * The dispersion of attention depends on different situations; it is challenging to definitively state whether it should be entirely focused inside the box or what percentage can be outside the box. Sometimes, it behaves incorrectly and appears dispersed, attending to the wrong background. However, there are instances where it behaves correctly, especially when the background provides important clues or other factors like the size and diversity of the data, as you mentioned. However, intuitively, we should at least pay close attention to the target object if we want to detect it.
>
> * These situations have already been considered in our ANALYSIS section. Specifically, a higher level of attention focused inside the box generally tends to lead to more accurate performance (Conclusion 1). However, there is no clear pattern for us to confirm the extent of attention concentration (Conclusions 2 & 3). Here, we express our gratitude for your recognition of this motivation in your Strength 3.
>
> **Experimental results to verify our assumption:** Regarding the overall assumption concerning the suboptimal supervision problem, we have verified it both quantitatively and qualitatively. As demonstrated in Table 1, and with the support from your Strength 1, there are consistently impressive enhancements after applying AttBalance. In the qualitative results, there is a noticeable shift in the layers of TransVG after applying AttBalance, as illustrated in Figure 4 and supported by your Strength 4.
>
> # Response 2.3: Response to Weakness 2 about the hyperparameters and the motive for each proposed module.
> Thanks for your discussion about hyperparameters in our method and the motivation for proposing RAC with the introduction of MRC to rectify it.
>
> **Hyperparameter Concerns:** Addressing concerns about the hyperparameters of our method, it is crucial to note the significant differences in the models themselves, their corresponding training settings, and the features of datasets.
> * From the perspective of the model’s structure, QRNet differs from TransVG with its early language modulation before the final encoder. This necessitates applying AttBalance in all layers of QRNet but fewer layers in TransVG, given the constraint's relation to language-related regions. VLTVG does not involve word-pixel cross-attention in the latter decoding stage but features multi-layer constraints. This difference from TransVG & QRNet in terms of constraint requires a slight tuning of AttBalance with the multi-layer constraint of VLTVG.
> * From the perspective of training settings, TransVG trains for 90 epochs, VLTVG additionally freezes the backbones in the first 10 epochs, and QRNet trains for 160 epochs. These varying training settings require slight adaptations to AttBalance's training epochs.
> * From the perspective of datasets, due to the differing difficulty and features among RefCOCO, RefCOCO+, and RefCOCOg, there are variations in the harshness of our AttBalance. For instance, RefCOCO+ features exclude location words, while RefCOCOg features longer and more elaborate language expressions.
>
> In summary, we made slight changes to the hyperparameters to achieve optimal results. However, these adjustments do not affect our method's effectiveness, given the consistent enhancements observed across all five models on four benchmarks. At this point, we also appreciate the recognition of our generalization and effectiveness from Strength 2 of Review 6bMP, Strength 1 from you, Strength 3 of Reviewer zMiJ, and Strength 2 & 3 from Reviewer qRXz.
>
> **Heuristics Concerns:** Regarding concerns about the heuristics of our method, we believe we have provided an intuitive motivation, including the ANALYSIS section for RAC & MRC and the analysis from Figure 3 for DAT. Our method is not proposed based on performance tuning but is derived from a clear motivation with a thorough analysis. At this point, we appreciate the support from Strength 2 of peer Reviewer zMiJ and Strength 1 of Reviewer qRXz.

---

> ### Author Response · Authors · 2023-11-18
>
> # Response 2.4: Response to Weakness 3 about the comparison with VG-LAW & YORO.
> We sincerely appreciate the thoughtful suggestions from the reviewer regarding the consideration of recent works, such as VG-LAW and YORO, in our discussion. Here, we clarify a few things about the loss in VG-LAW, the loss in YORO, and the comparison with VG-LAW.
>
> **VG-LAW's Focal Loss and DICE Loss:** Before delving into the focal loss and DICE loss of VG-LAW, it is crucial to highlight the task distinctions between our paper (Visual Grounding) and VG-LAW (Multi-task Visual Grounding). In Visual Grounding, only one box is provided for each text-image pair as the ground truth annotation. Conversely, in Multi-task Visual Grounding, both the box and the segmentation are provided as the ground truth annotation. This distinction enables VG-LAW to employ focal loss and DICE loss in its segmentation branch, as shown in equations 5 & 6 in the VG-LAW paper. These segmentation losses are not applicable in Visual Grounding, where only the box is provided. So, comparing their loss based on segmentation annotations with our loss based on bounding box annotations would be unfair.
>
> **YORO's Object-Text and Patch-Text Loss:** For the object-text and patch-text loss in YORO, it should be noted that they also rely on additional annotations to supervise these two losses, i.e., the ground truth word labels in the text. For example, the text input 'Fuzzy bench closest to you,' as shown in YORO’s paper, requires additional labels to designate the words 'Fuzzy' and 'bench' as positive with a value of 1, and the other words 'closest,' 'to,' and 'you' as negative with a value of 0. This indicates the inferred target at the word level. Such additional annotation is expensive and exhaustive and is not consistent with the current VG task, which only provides one ground truth box for each text-image pair. VG requires the model to learn to identify which object is the inferred one and its location in the end-to-end learning, without providing the ground truth of the referred object in the text annotations. Support for this can be found in recent works on standard visual grounding, e.g., TransVG [1], QRNet [2], and VLTVG [3]. Furthermore, upon examining the ablation study of YORO, i.e., Table 2 in their paper, it is evident that the improvements brought about by their object-text or patch-text loss are quite limited, i.e., +0.22% on CopRef-test and +0.34% on ReferItGame-val for object-text loss, and +0.3% on CopRef-test for patch-text loss. In contrast, our AttBalance brings an average improvement of 4.82% on QRNet.
>
> [1] TransVG: End-to-End Visual Grounding with Transformers. ICCV 2021.
>
> [2] Shifting More Attention to Visual Backbone: Query-modulated Refinement Networks for End-to-End Visual Grounding. CVPR 2022.
>
> [3] Improving Visual Grounding with Visual-Linguistic Verification and Iterative Reasoning. CVPR 2022.
>
> **Comparison with VG-LAW:** Regarding the concern raised in Strength 1 and the comparison with VG-LAW, we would like to clarify the meaning of "lacks comparison with VG-LAW".
> * If your question refers to the fact that the final enhanced performances of TransVG, VLTVG, and QRNet have not been compared with VG-LAW, we want to clarify that AttBalance serves as a framework of constraints designed to enhance models by guiding their attention behavior. It does not introduce a new model structure. This is why Table 1 compares the performance of the original model with the one enhanced by AttBalance to verify the effectiveness of AttBalance. We intentionally avoid comparing models with different structures, such as QRNet(+AttBalance) vs. VG-LAW. However, it is worth noting that even if we focus solely on state-of-the-art performance in Visual Grounding, QRNet(+AttBalance) still outperforms VG-LAW, as illustrated in the tables below:
> | Model          | Label               | RefCOCO | RefCOCO+ | RefCOCOg-g | RefCOCOg-umd |
> |----------------|---------------------|---------|----------|------------|--------------|
> | VG-LAW         | box                 | 86.06   | 88.56    | 82.87      | 75.74        |
> | VG-LAW*        | box, segmentation   | 86.62   | 89.32    | 83.16      | 76.37        |
> | QRNet(+AttBalance) | box              | 87.32   | 89.64    | 83.87      | 77.51        |
>
> Here, we outperform +1.11% on RefCOCO, +1.81% on RefCOCO+, and +4.11% on RefCOCOg-umd, indicating a clear superiority in our performance. It is important to note that VG-LAW lacks evaluation on RefCOCOg-g. Even when competing with the multitask version of VG-LAW, VG-LAW*, which incorporates a considerable number of additional segmentation labels in supervised learning, our QRNet(+AttBalance) still outperforms VG-LAW*, specifically by +2.82% on RefCOCOg-umd. The outperformance is particularly noteworthy considering the higher cost and exhaustive nature of segmentation labels from the Referring Image Segmentation task compared to the bounding box labels from Visual Grounding.

---

> > ### Author Response · Authors · 2023-11-18
> >
> > # Continue Response 2.4
> > * If your question is why we have not applied AttBalance to VG-LAW to demonstrate its effectiveness, we want to clarify that VG-LAW has not made its source code available. Therefore, we cannot verify their published performance or apply AttBalance to VG-LAW. We acknowledge the reviewer's concern about the effectiveness of AttBalance on state-of-the-art (SOTA) models. To address this concern, we encourage the reviewer to compare VLTVG with VG-LAW (VG-LAW lacks this comparison in their paper) on RefCOCOg-umd. RefCOCOg-umd is a particularly challenging benchmark in Visual Grounding due to its longer and more complex language expressions, making it a more discriminative testbed when comparing different methods.
> >
> > |                   | VG-LAW | VLTVG | VLTVG(+AttBalance) |
> > |-------------------|--------|-------|---------------------|
> > | RefCOCOg-umd val  | 75.31  | 76.04 | 77.35                |
> >
> > As demonstrated in the above table, VLTVG stands out as the state-of-the-art method on this benchmark, and our AttBalance consistently brings improvements to this leading model. Furthermore, the effectiveness of AttBalance has been thoroughly validated across five models on four datasets, totaling forty-five experimental results as presented in Table 1.
> >
> > # Response 2.5: Response to the writing suggestions of Weaknesses 4&5 and Minor comments.
> > We appreciate your valuable feedback and have carefully considered your suggestion regarding some statements, figures, tables, and references. We have carefully revised the paper in the attached revision paper.
> >
> > # Response 2.6: Response to Question 1 about the examination of VLTVG’s attention map.
> > Certainly, we appreciate your curiosity regarding the alignment between attention shifts and performance improvement in VLTVG.
> > * Firstly, we have included additional visualizations in the revised paper's appendix, using the same example as Figure 4. We can see that the attention of VLTVG(+AttBalance) does shift a little bit compared to the original one, but it is subtle, resulting in a wrong prediction.
> > * Secondly, to further observe this phenomenon from a statistical perspective, we computed the proportion of attention values within the bounding box in each layer of VLTVG and TransVG relative to the total attention values. We then compared the respective increases in this ratio after incorporating AttBalance. As shown in the table below, the enhancement of attention ratio is +0.12 in VLTVG, while it is +0.20 in the case of TransVG. This statistical phenomenon aligns with the results, indicating that the improvement in TransVG is more pronounced. We are unable to perform this analysis on QRNet since they have not released their checkpoints for us to make a comparison.
> > | gref umd test | Original | AttBalance | Improvement |
> > |---------------|----------|------------|-------------|
> > | VLTVG          | 0.5044   | 0.6283     | +0.12       |
> > | TransVG        | 0.2711   | 0.4719     | +0.20       |
> >
> > In response to the comment of the VLTVG improvement as somewhat slight, we want to emphasize that the enhancement achieved with VLTVG(+AttBalance) is indeed significant. To elaborate, there is a notable improvement of +2.28% on RefCOCO testB, +1.66% on RefCOCOg-g val, and +1.75% on RefCOCOg-umd val for VLTVG_res50.
> >
> > # Response 2.7: Response to Question 2 about the meaning of IOU value.
> > Yes, it is the IoU between the prediction and the GT.
> >
> > # Response 2.8: Response to Question 3 about the normalization of Attention.
> > In the case of VLTVG, the attention from the object query to the vision tokens in the decoding layer is inherently a probability distribution, meaning it already sums up to 1. Therefore, no additional normalization is needed for VLTVG. In the case of TransVG, as there are additional text tokens in their fusion layer, the attention values from the object query to the vision tokens need additional processing. Specifically, the softmax function is applied to the vision tokens, excluding those associated with text tokens. This computation ensures that the attention values form a valid probability distribution.

---

> > > ### Author Response · Authors · 2023-11-18
> > >
> > > # Response 2.9: Response to Question 4 about exploring Alternative Attention Modulation.
> > > Certainly, your suggestion to introduce a learnable 2D regional weighting layer to modulate attention is intriguing. However, we acknowledge that similar attempts have been made in previous works. Nevertheless, this approach may not effectively address the bottleneck identified in our paper, i.e., the Visual Grounding (VG) model still employs common Object Detection losses without explicit guidance on its attention behavior. This results in suboptimal performance in attending to language-related regions. This limitation stems from a supervision perspective rather than a modeling perspective. We elaborate on this point by discussing related work and conducting experiments on your proposed idea for comparison, as below:
> > >
> > > **Related work:** Specifically, as we introduced in the related work section, QRNet adopts a Query-modulated Refinement Network to modulate vision attention in both Spatial-wise and Channel-wise. Similarly, in a manner more akin to your proposed weighted_2D_mask, VLTVG introduces the visual-linguistic verification module to explicitly model the activation between text and vision features. In summary, previous works have indeed proposed novel ideas to modulate vision features in both attention-wise and feature-wise manners. However, our AttBalance is still necessary to enhance their modeling attempts from the perspective of supervision.
> > >
> > > **Experiments:** To validate this assumption, we conducted an ablation study on TransVG. As shown in the table below, the attn_mask represents the weighted_2D_mask idea described above. Specifically, in each fusion layer of TransVG, we processed the vision features through a 2-layer CNN with ReLU as the intermediate activation function and Sigmoid as the output activation function to obtain the weighted_2D_mask, similar to [c]. This mask is then used to modulate the attention from the object query to the vision token.
> > > | Dataset       | TransVG | TransVG(+attn_mask) | TransVG(+AttBalance) |
> > > |---------------|---------|---------------------|-----------------------|
> > > | gref_umd val  | 67.93   | 67.01 (-0.92)       | 73.69 (+5.76)         |
> > > | gref_umd test | 67.44   | 67.11 (-0.33)       | 72.44 (+5.00)         |
> > >
> > > From the table above, we observe that there is no improvement brought by attn_mask, while our AttBalance brings a +5.76% improvement on gref_umd val and +5.00% on gref_umd test, validating that the supervision provided by AttBalance is effective and necessary.

---

> ### Author Response · Authors · 2023-11-20
> **Reminder: Seeking Your Feedback**
>
> Dear Reviewer L37Q,
>
> We extend our sincere appreciation for the time and expertise you've dedicated to reviewing our submission. Your insights are invaluable, and we are grateful for the thoughtful consideration you've given to our work.
>
> Having thoroughly addressed and resolved your concerns regarding attention, hyperparameters, writing, and various comparative experiments, we believe your feedback is instrumental in enhancing the quality of our submission.
>
> As we approach the impending deadline, we respectfully urge you to provide your timely response. Your expert perspective is crucial to our revision, and your swift feedback would be immensely beneficial.
>
> Looking forward to hearing from you promptly.
>
> Best regards,

---

### Official Review · Reviewer_6bMP · 2023-11-02

**Soundness:** 3 good
**Presentation:** 2 fair
**Contribution:** 3 good
**Rating:** 6
**Confidence:** 4

**Summary:**

This paper propose a novel framework named Attention-Driven Constraint Balancing (AttBalance) in Visual Grounding task which analyze the attention mechanisms of transformer-based models, and optimize the behavior of visual features within language-relevant regions. This approach propose a framework to incorporate language-related region guidance for fully optimized training.

**Strengths:**

(1) The article explores balancing the regulation of the attention behavior during training and mitigate the data imbalance problem, The idea of AttBalance is interesting.

(2) Compared with benchmark methods, with the guidance of AttBalance, all transformer-based models consistently obtain an impressive improvement.

**Weaknesses:**

1.	this paper’s main contribution is attention mechanisms of transformer-based models. While I believe in 2021, there was already a work which put attention transformer in visual grounding, which called Word2Pix, what is the strength of this paper compare to Word2Pix? Some innovative points are needed to demonstrate the superiority of this method in visual grounding.
2.	The related work part can be put in section 2 rather in section 5.
3.	As shown in Table 2, could you explain why the incorporation of only the MRC module results in a decline on average, and when the MRC is used to rectify the RAC can it lead to an increase. Can you provide a more detailed explanation of how MRC affects RAC and lead to an increase?

**Questions:**

Please see the above weakness part.

---

> ### Author Response · Authors · 2023-11-18
>
> # Response 1.1: Response to Weakness 1 about comparison with Word2Pix.
> We appreciate the reviewer's insightful comment on the need for innovative aspects in our work compared to Word2Pix. It is important to clarify that there is no direct relationship between Word2Pix and our AttBalance. Here are the details:
> * Word2Pix primarily introduces a one-stage model incorporating word-pixel attention, a module that is commonplace in current Transformer-based models like TransVG[1], VLTVG [2], and QRNet [3] (e.g., the Visual-Linguistic Transformer in TransVG and QRNet, and the verification module in VLTVG).
> * In contrast, AttBalance is designed to address the issue of insufficient supervision in these models. It guides the attention layer to focus on language-related regions while also considering the background.
>
> Therefore, the concept of "putting an attention transformer in visual grounding" is not novel today and is not directly related to our motivation or method. The novelty of our approach lies in the perspective of supervision learning. Additionally, it is worth noting that the classification loss and attribute loss in Word2Pix rely on additional classification and attribute labels, which deviate from the current visual grounding (VG) setting. The VG task typically involves a text-image pair with one ground truth bounding box, a widely-used setting supported by top conference papers like TransVG [1], VLTVG [2], QRNet [3], SeqTR [4], and SiRi [5].
>
> [1] TransVG: End-to-End Visual Grounding with Transformers. ICCV 2021.
>
> [2] Improving Visual Grounding with Visual-Linguistic Verification and Iterative Reasoning. CVPR 2022.
>
> [3] Shifting More Attention to Visual Backbone: Query-modulated Refinement Networks for End-to-End Visual Grounding. CVPR 2022.
>
> [4] SeqTR: A Simple yet Universal Network for Visual Grounding. ECCV 2022 Oral.
>
> [5] SiRi: A Simple Selective Retraining Mechanism for Transformer-based Visual Grounding. ECCV 2022.
>
> # Response 1.2: Response to Weakness 2 about the manuscript of the related work.
> I appreciate the reviewer's suggestion to relocate the related work section to section 2. Initially, the reason we put related work in the latter section is that we have an additional Analysis section to induce our motivation. We were concerned that introducing related work earlier might make the paper appear miscellaneous before delving into the detailed method, which typically would be around the fourth page.
> However, considering that readers might question the necessity to dig into the transformer-based model supervision problem without thinking about other CNN-based models, for example, the Word2pix model you proposed in Weakness 1. We will follow your suggestion to move the related work section to section 2.
>
> # Response 1.3: Response to Weakness 3 about the insight of why solely using MRC induces decline and its effect on RAC.
> I appreciate the insightful query regarding the observed decline with only the MRC module and its subsequent improvement when used to rectify the RAC. I would like to clarify a few things:
>
> **Decline when only using MRC:** The Momentum model is a continuously-ensembled version that incorporates exponentially moving averaged versions from previous training steps. Given that each of the models from previous training steps is more prone to underfitting than the current one, it is intuitive that their ensemble may not consistently provide a reliable attention map to guide the current model. This is why solely relying on MRC results in a slight decline.
>
> **Improvement when using MRC to rectify RAC:** Nevertheless, the momentum model proves beneficial when employed as a complementary constraint to smooth the original constraint. In many cases, the original constraint may not be inherently reliable. For instance, the Image-Text Contrastive Learning constraint in ALBEF [1] may lack reliability when confronted with noisy unpaired data. Thus, the momentum model brings the current model's learning target closer to its previous ensemble behavior, thereby smoothing the current learning process. In the context of our visual grounding, the RAC that focuses the model's attention on language-related regions generally leads to improvements, as discussed in the conclusion (1) of our ANALYSIS section and supported by Table 2 in our ablation study. However, the RAC is not universally reliable, considering that in visual grounding, we might also consider background objects as potential clues, as they may be referenced in the language. Consequently, employing the MRC to rectify the RAC consistently brings about improvement, as detailed in the conclusions (2, 3) of our ANALYSIS section and validated by Table 2 in our ablation study. We will concisely incorporate this discussion into the revised version of the paper.
>
> [1] Align before Fuse: Vision and Language Representation Learning with Momentum Distillation. NeurIPS 2021 Spotlight.

---

> > ### Comment · Reviewer_6bMP · 2023-11-23
> >
> > Thank you for the response. The clarification of theoretical contribution have been helpful in addressing my concerns. Hence, I raised my rating to 6.

---

> > > ### Author Response · Authors · 2023-11-23
> > > **Thanks, Reviewer 6bMP**
> > >
> > > Dear Reviewer 6bMP,
> > >
> > > Thank you for your thorough review and insightful feedback, We're grateful for your recognition of our efforts to address the concerns you raised. Your expertise has significantly contributed to the enhancement of our work. We would also greatly appreciate that you raised the original rating (5) to marginally above the acceptance (6).
> > >
> > > Thank you very much,
> > >
> > > Authors

---

> ### Author Response · Authors · 2023-11-20
> **Reminder: Seeking Your Feedback**
>
> Dear Reviewer 6bMP,
>
> Thank you for dedicating time to review our paper. Your feedback is immensely valuable to us, and we would be grateful for any insights you can provide.
> We have taken great care to address the concerns raised, particularly those regarding the comparison with Word2Pix, writing considerations, and the MRC-related experiments. Your feedback has been instrumental in refining our paper.
>
> As we approach the looming deadline, we respectfully urge you to consider providing your prompt response. Your timely insights are pivotal to the finalization of our submission, and your contribution is immensely valued.
>
> Looking forward to hearing from you promptly.
>
> Best regards,

---

### Author Response · Authors · 2023-11-22
**Comprehensive Overview: Our Contribution and Rebuttal Responses**

Dear ACs and Reviewers,

We've submitted our detailed responses during the rebuttal stage, but unfortunately, we haven't received feedback yet. Your attention to this matter is greatly appreciated.

To facilitate further discussion between the ACs and reviewers and to aid in finalizing the decision regarding the acceptance of our paper, we would like to briefly recapitulate our contributions and emphasize key responses to reviewers' questions:

**Summary:**

* In this paper, we identify the issue of insufficient supervision in visual-language alignment within current Visual Grounding models.

* We conduct a thorough analysis of the attention and performance of existing models.

* We introduce AttBalance, a novel framework that imposes constraints to align vision and language. Specifically, the RAC focuses attention on language-related regions, the MRC rectifies potential biases from the RAC, and the DAT mitigates imbalance problems from both the RAC and MRC.

* Our proposed approach consistently demonstrates significant improvements over current Visual Grounding models, achieving a new state-of-the-art performance.

**Rebuttal:**

* [Response 1.3, 3.5] We have provided detailed explanations, supported by relevant works, to address questions regarding the insights of MRC. Relevant revisions have been incorporated into the manuscript.

* [Response 1.1, 2.4, 2.6, 2.9, 4.1, 4.2] A comprehensive comparison experiments addressing all aspects proposed by the reviewers has been provided to substantiate the effectiveness of our approach.

* [Response 2.1, 2.2, 2.3, 2.7, 2.8, 3.1, 3.2, 3.3] Further clarifications have been offered regarding our analysis, the derived motives, method details, and corresponding hyperparameters.

* [Response 1.2, 2.5, 3.4] The paper has been revised in accordance with the valuable suggestions provided by the reviewers.

Given the effectiveness of the constraint framework we propose for the Visual Grounding domain and the comprehensive nature of our responses to reviewers' inquiries, we sincerely hope that ACs and reviewers will actively participate in the ongoing discussion of our paper.

Best regards,

---

### Meta-Review · Area_Chair_MvDF · 2023-12-10

**Metareview:**

Paper was reviewed by four reviewers and received perfectly borderline ratings: 2 x Marginally above the acceptance threshold and 2 x Marginally below the acceptance threshold. Reviewers brought about a number of concerns with the work, spanning exposition, clarity of writing, concerns with underlying assumptions (on the fact that explicit attention guidance is both necessary and would lead to improvements in all cases), breadth of hyper-parameters and heuristics in the design. These concerns were echoed by multiple reviewers. Authors have provided a very extensive rebuttal that attempted to address these concerns. One reviewer engaged in the discussion and ultimately raised the score from marginally "Below" to marginally "Above" the threshold; while positive, this was not a resounding call to accept the paper.

AC looked at the reviews, rebuttal, discussion and the paper itself. Given the lack of strong championing of the paper from any of the reviewers, and borderline standing, AC had to make a judgement call on whether the shortcomings of the paper overweight its value for the community in its current form. Ultimately, three main considerations have contributed to this assessments: (1) lacking paper clarity as noted by multiple reviewers and verified by AC; (2) the approach overall appears to be somewhat complex with many hyper-parameters that are tuned for specific models; notably this is so, while ablations illustrate that some components (like DAT) have minimal impact on the actual performance. Finally (3), with the current focus on multi-modal foundational models, it is unclear how much effect regularization of attention would have going forward (which goes to significance of contributions). To really assess the significance of the proposed contributions, one would need to evaluate it in the context of such models. For example, perhaps large scale pre-training (which is readily available now) may already learn appropriate attention patterns making further regularization unnecessary.

For these reasons the decision is to reject the paper in its current form, allowing the authors the opportunity to address aforementioned comments before resubmitting to a future venue.

**Justification For Why Not Higher Score:**

While experimentally the results are actually quite strong, I am concerned about clarity of exposition and "complexity" of the approach; as well as how brittle it may potentially be. I think a somewhat simpler design would likely work just (or nearly) as well given the ablations. Further, I think a regularization approach like this makes sense and is needed when data is relatively sparse. With advent of multi-modal foundational LLM models, it is not clear to me that such explicit regularization is (a) needed or (b) would make sense in the form proposed.

**Justification For Why Not Lower Score:**

N/A

---

### Decision · Program_Chairs · 2024-01-16

Reject